# The Effects of Housing Environments on the Performance of Activity-Recognition Systems Using Wi-Fi Channel State Information: An Exploratory Study

**DOI:** 10.3390/s19050983

**Published:** 2019-02-26

**Authors:** Hoonyong Lee, Changbum R. Ahn, Nakjung Choi, Toseung Kim, Hyunsoo Lee

**Affiliations:** 1Department of Architecture, College of Architecture, Texas A&M University, College Station, TX 77843-3137, USA; onarcher@tamu.edu; 2Department of Construction Science, College of Architecture, Texas A&M University, College Station, TX 77843-3137, USA; 3Nokia Bell Labs, Murray Hill, NJ 07974-0636, USA; nakjung.choi@nokia-bell-labs.com; 4Department of Architecture and Architectural Engineering, Seoul National University, Seoul 08826, Korea; toeskim92@gmail.com (T.K.); hyunslee@snu.ac.kr (H.L.)

**Keywords:** smart home, occupant activity recognition, channel state information (CSI), Wi-Fi, housing environment

## Abstract

Recently, device-free human activity–monitoring systems using commercial Wi-Fi devices have demonstrated a great potential to support smart home environments. These systems exploit Channel State Information (CSI), which represents how human activities–based environmental changes affect the Wi-Fi signals propagating through physical space. However, given that Wi-Fi signals either penetrate through an obstacle or are reflected by the obstacle, there is a high chance that the housing environment would have a great impact on the performance of a CSI-based activity-recognition system. In this context, this paper examines whether and to what extent housing environment affects the performance of the CSI-based activity recognition systems. Activities in daily living (ADL)–recognition systems were implemented in two typical housing environments representative of the United States and South Korea: a wood-frame apartment (Unit A) and a reinforced concrete-frame apartment (Unit B), respectively. The experimental results show that housing environments, combined with various environmental factors (i.e., structural building materials, surrounding Wi-Fi interference, housing layout, and population density), generate a significant difference in the accuracy of the applied CSI-based ADL-recognition systems. This outcome provides insights into how such ADL systems should be configured for various home environments.

## 1. Introduction

In order to advance smart home environments that can deliver elderly healthcare, energy savings, and home security, systems must first be able to accurately recognize human activity in daily living (ADL) [1,2]. Traditionally, ADL-recognition systems rely on dedicated sensors such as cameras, motion sensors, or other special sensors (e.g., inertial measurement units). However, these device-based ADL-recognition solutions have limitations in their use, especially since these systems require significant infrastructural installations in the environment—for example, some such systems require cameras or motion sensors to be attached to walls or doors to detect activity. Problematically, some of these technologies face inherent limitations—e.g., cameras raise privacy issues and require line-of-sight for human movements—which makes installation concerns and considerations more pressing. Alternatively, wearable, sensor-based approaches that require users to wear the sensors to detect activities may offset the infrastructural concerns, but these wearable approaches demand users’ diligent applications of the devices, a fact that can challenge the effectiveness of the technology [1,2,3]. For these reasons, ADL-recognition still represents a puzzle to many smart-environment developers.

In recent years, device-free activity-recognition approaches have been the focus of ADL-recognition systems [2], with Wi-Fi signal systems presenting an especially attractive option. Due to its ubiquitous presence in home environments, Wi-Fi signals have already been employed for human-activity recognition without additional devices. Such Wi-Fi signal-based approaches do not require a dense placement of sensors to generate detecting areas of interest. These systems consist of a Wi-Fi transmitter (an Access Point, AP) and one or several Wi-Fi devices (Receivers) located in different places within the environment. Also, these Wi-Fi signal-based ADL-recognition systems do not require human activity in the Line-of-Sight (LOS) or face privacy problems [4], making them a reasonable alternative to device-based ADL-recognition systems.

Wi-Fi signal-based ADL-recognition approaches exploit fine-grained Wi-Fi signal signatures. Specifically, Wi-Fi signals propagate from a transmitter to receivers, so human activity may affect the signals’ propagation paths, which in turn cause the signals to be changed at receivers. For example, Received Signal Strength (RSS) has been used for fingerprint-based localization systems [5,6,7,8,9,10,11,12]; when a subject is located between the AP and the receiver, the RSS is changed due to signal attenuation. Exploiting this RSS, coarse human activity (e.g., vacant home, occupied home, human movement, walking activity) has been detected with an average 90% accuracy [13]. However, the option to exploit RSS for ADL-recognition is only available for coarse-level activity detection because RSS captures total power and exhibits signal variance as a single amplitude, rendering the approach ineffective in a static and/or complex environment [14,15].

Unlike RSS which has a superimposition layer of multipath signals, Channel State Information (CSI) exploits channel information between the AP and the receiver at the individual subcarrier level. Specifically, CSI represents a channel response to the physical environmental changes, which depicts the multipath effects of signals. Consequently, CSI is more stable and robust than RSS, and CSI data can be captured from commodity Wi-Fi devices using a Linux CSI 802.11n tool [16]. Recently, CSI-based ADL-recognition approaches have succeeded at detecting human activities at different levels of granularity ranging from coarse to fine, as evidenced by the detection successes of technologies such as Wi-Sleep [17], Wi-Chase [15], E-eyes [18], and RT-Fall (Real-Time Fall) [3]. These approaches can not only measure various daily activities, but also the fine movement of the chest during breathing in real time.

Despite these advantages, these CSI-based ADL-recognition systems environments still need further verification in terms of their performance in various real-world housing environments; Most previous studies [15,19,20,21,22] were conducted in controlled laboratory settings and several few attempts in real-world environments were mainly conducted in a representative housing environment in the United States [3,18]. The Housing environment greatly varies by its location and housing type, and the difference of the housing environment may create a significant impact on the performance of CSI-based ADL-recognition. For example, Wi-Fi signals have different propagation loss as Wi-Fi signals penetrate different building materials, such as wood, glass, and concrete [23,24]. In addition to building materials, other factors such as unit layouts and distance to neighbor units would affect the performance of CSI-based ADL-recognition. However, it has not been investigated whether and to what extent existing CSI-based ADL recognition systems can be resistant to different housing environments. To this end, this paper examines the effects of housing environments on the performance of CSI-based ADL-recognition systems. In particular, this study selected two units that provide a great difference in their housing environments, including structural and finish materials, and unit density, and investigated whether such a difference in housing environment creates a noticeable difference in the performance of CSI-based ADL-recognition, regardless of the algorithms used for ADL recognition.

## 2. Related Work

Recently, exploiting Wi-Fi signals for activity detection has risen in popularity due to the availability and ubiquitous distribution of these wireless networks and their corresponding commercial devices. Human-body movements cause Wi-Fi signals to change when the signals are reflected from the body, which results in variations at the Wi-Fi receiver and, consequently, opportunities to estimate human activities by analyzing the signal variance. This approach further benefits from the fact that Wi-Fi signal–based ADL-recognition systems do not require LOS for the activities since Wi-Fi signals propagate through walls. Thus, systems such as Wi-Vi [25] and WiSee [26] have successfully detected the existence of humans and have differentiated several human gestures—such as punch, kick, and push—even on the other side of a wall.

While such RSS approaches have been successfully used for gross-level ADL recognition, CSI manifests even greater sensitivity to fine-activity differentiation. CSI is more stable and robust than RSS [20], and the approach can be accessed using several off-the-shelf Wi-Fi devices (e.g., Intel Wi-Fi Link 5300 NICs and Atheros AR9580 NICs). Furthermore, CSI can be successfully extracted from Wi-Fi signals using a readily available CSI 802.11n tool [16].

To-date, CSI-based ADL-recognition systems have been applied to various activity-detection tasks. Firstly, CSI has been used for indoor localization. Location detection indoors is required in various settings, such as in hospitals (patient tracking) and disaster areas (personnel locating). Wu et al. [27] compared the accuracies between an RSS-based indoor localization solution and a CSI-based indoor localization solution using probabilistic approaches (which provide more accuracy than deterministic approaches [28]); deep learning techniques were employed to reduce the location error of the CSI-based indoor location systems [20]. In order to reduce complexity, time or system processes, Wu et al. [27] divided their system into two states: an offline process and an online process. The offline process served as the training stage for database construction, which in turn trained the CSI fingerprinting function; then, the online process recorded real-time data and tested the CSI-based approach by exploiting the database. Their results outperformed existing indoor localization systems.

Another CSI-based approach, Wi-Chase [15], recognized coarse activities—such as walking, running, and moving hands—by exploiting all CSI-subcarrier data. This approach used two machine-learning algorithms, k-Nearest Neighbor (kNN) and Support Vector Machine (SVM), and obtained the highest accuracy for hands moving because hands moving has similar repeating patterns in a fixed position, unlike locomotive activities. The Wi-Chase study also showed that when more subcarriers were used with multiple AP and receiver links, the performance of the system improved.

For more diverse in-place activities and for walking-direction recognition, Wang et al. [18] proposed the E-eyes algorithm. Unlike Wi-Chase, E-eyes algorithm selected known activity data and measured the similarity between the known activity data and the unknown activity data to recognize the unknown activity. Specifically, the E-eyes algorithm first differentiated between walking activity and in-place activity using CSI variance, since walking activity causes higher variance in CSI than in-place activity. Then, in-place activities were estimated based on their similarity with known activities using Earth Mover’s Distance (EMD), and walking directions were detected by Dynamic Time Warping (DTW). The results of the Wang et al. [18] study showed that higher packet-transmission rates yielded higher accuracy in activity recognition.

Building upon these successes, researchers have applied CSI-based ADL-recognition solutions to various smart home healthcare systems. Wang et al. [3] proposed RT-Fall algorithms to detect falls, since fall detection is essential for elderly healthcare in a smart home. The process of detecting falls using CSI requires high packet-transmission rates and small window sizes because falling occurs in a very short time [3]. In general, the frequencies of fall and fall-like activities lie between 5 and 10 Hz, while in-place activities lie in the lower frequency range, from 0 to 4 Hz. RT-Fall analyzed the frequency of CSI to recognize fall and fall-like activities such as sitting down and lying down. They recognized falls in real time with approximately 90% accuracy. Borhani and Pätzold [29] developed a simulation model for the Wi-Fi-based fall detection system using a stochastic 3D trajectory model. In the simulation, human body, such as the head, arms, and legs, are molded as moving scatters, and fixed scatters represents static objects (e.g., the walls, appliances, or furniture). The simulation model detects when a fall occurs during random walking by analyzing the time-variant Doppler effect caused by an occupant’s activity. 

In keeping with the healthcare applications, CSI has been exploited to detect vital signs such as respiration and heartbeat rates. PhaseBeat [30] used CSI phase data to detect vital signs because phase data are more stable than amplitude and manifest periodicity. In order to detecting heart rates, the researchers used a directional antenna at the AP to improve the reflected signal power—heart movements are too weak to cause variance in the reflected signals. Similarly, Liu et al. [17] detected breathing rates using CSI. In their study, an AP and receiver were placed at two sides of the subject for better signal quality. They noticed that sleeping positions affected the performance of respiration detection: If a person is in ‘Fetus,’ ‘Log,’ or ‘Yearner’ sleeping positions, the back of the subject blocks the Wi-Fi signals’ paths. Thus, the researchers determined that users should change the location of the AP-Receiver pair to detect chest movement.

CSI-based ADL-recognition systems have also been used in place of human-device interactions. Various smart home gadgets control such home appliances as TVs, laptops, and mobile phones. Alternatively, Nandakumar et al. [31] used CSI to control these home appliances by detecting human gestures; they obtained an average 91% and 89% accuracies when the receiver was located in LOS and in a bag, respectively. In another study, Ali et al. [4] focused on keystroke recognition using CSI. When a person types a specific key, his hands and fingers move in a unique pattern; however, the movements of hands and fingers are micro-movements and some unique patterns for different keys are almost identical—for example, ‘F’ and ‘G’ keys are closely placed and may easily be confused. In order to solve such nuance, the researchers extracted features from the shapes of keystroke waveforms instead of from the CSI values themselves, since the CSI values of many keys have similar features—such as maximum value, mean value, or root mean square deviation—but have different waveforms, and the shapes contain both a time and a frequency domain. Their WiKey algorithm obtained an approximately 94% keystroke-recognition accuracy.

These successfully developed and verified CSI-based ADL-recognition systems benefit from CSI’s stability and accessibility. However, Wi-Fi signals are affected by various environmental factors, which can in turn have impact on the performance of CSI-based ADL-recognition systems. Thus, in order to advance the opportunities for applying CSI to human ADL recognition in smart homes, studies must verify that these systems’ performances will yield consistent accuracy in different housing environments.

## 3. Background

As Wi-Fi signals propagate in physical space, the signals reach receivers (Wi-Fi devices) through various routes, a concept known as multipath. Figure 1 shows the multipath of signal propagation: The received signal is composed of signals arriving over many different paths, all which can be affected by environmental factors [3]. The environmental factors therefore combine with scattering, fading, and power decay over distances, and these environmental effects on the signals can be manifested in the CSI [18]. For example, while CSI remains stable in a static home environment, if a person performs activities, Wi-Fi signals scatter in response to the body’s movement, thereby causing bistatic Doppler shift at the receiver. The bistatic Doppler frequency depends on the occupant’s moving speed, the Wi-Fi frequency band, and the relative position between the occupant and the Wi-Fi transceivers [32].

Mathematically, the CSI matrix, $H_{i}$, is related to the transmitted signal vector $X_{i}$ and the received signal vector $Y_{i}$, as shown in Equation (1) [15].
(1)$$Y_{i}=H_{i}X_{i}+{Ɲ}_{i}$$
where i is the data packet, i∈[1, N]; N is the number of received packets; $Y_{i}$ is the received signal vector; $H_{i}$ is the CSI matrix; $X_{i}$ is the signal vector; and ${Ɲ}_{i}$ is the noise vector.

CSI has several subcarriers that are divided by Orthogonal Frequency Division Multiplexing (OFDM) [2]. Thus, the CSI matrix for a packet, H, has 30 subcarriers, each with three transmission antennas and three receiver antennas. Hence, the total number of information pathways for a sending packet is 270 CSI amplitude and another CSI 270 phase. Unlike RSS, which only has one path per packet, CSI exploits multiple subcarriers that travel along different fading or scattering multipaths [27] in order to better denote data across dimensions of time and space.

Figure 2 shows the raw CSI amplitude data of 30 subcarriers for a walking activity. The different colors indicate 30 subcarriers that have different amplitude values but show a similar tendency. When a subject is standing, the CSI is relatively stable. However, the amplitudes fluctuate as the subject starts to walk. The black area in Figure 2—which indicates when the walking activity occurred—has relatively high amplitude variance. CSI-based ADL-recognition approaches exploit such CSI pattern changes [18] to identify human activity.

## 4. Methodology

In order to examine whether and how CSI-based ADL-recognition systems can be resistant to varying housing environment, two different housing environments were selected: A wood-frame, low-rise apartment in the United States (Unit A), and a reinforced concrete-frame, high-rise apartment in South Korea (Unit B). These two housing environments show clear differences in construction materials and population densities. The exterior walls in Unit A were composed of wood framing and insulation, and wood sheathing, and its interior walls were also composed of wood framing and drywalls with painting finishes. Unit B was built with reinforced concrete structure and its interior walls were masonry and drywall, and wallpapers were used for the interior wall finishing. While Unit A is also a part of low-rise multi-family housing, the population density of the high-rise apartment complex that Unit B belongs to is much higher. The layouts of two units present clear differences, as shown in Figure 3. However, the sizes of two units in terms of floor space are quite similar, a Wi-Fi router and a receiver were located similarly in each unit (in living room), and the distance between the router and the receiver was also quite similar. Because the two units were actual living spaces, there were miscellaneous household items. The type, number, and location of household items were different in the two units. Both Unit A and Unit B contains large household items: a refrigerator in the kitchen, a dining table in the dining room, and a desk in the living room. However, there were more small and medium sized household items (e.g., furniture, appliances) in Unit B than in Unit A. Four different activities, including walking, eating, typing, and no-activity, were recognized by the CSI-based systems. Two algorithms were used in ADL recognition in order to analyze the housing environment effect independent from algorithms. The ADL-recognition using these algorithms followed the four steps: (1) Data collection, (2) Data preprocessing, (3) Activity segmentation, and (4) Activity classification. This section discusses the experimental setup, the two activity detection approaches, and how they were compared.

### 4.1. Data Collection

Two subjects were recruited for the experiments at each Unit; Recruiting for the experiment at each Unit was conducted separately due to the geographical distance. All the four subjects participating in both Units were male having similar physical characteristics; Their heights ranged from 175 cm to 180 cm, while their weights range from 68 kg to 70 kg. During the experiments, the subjects were instructed to perform the identical activities, and tests were performed by one subject at a time. Each subject walked in ten rounds, as indicated by the arrows, and performed eating and typing in ten rounds in the dining room and living room, respectively. Each round required 10 s of activity and 20 s of no-activity. The 20 s interval between activities clearly differentiated the multiple activity rounds. Figure 3 shows the walking trajectories in the test beds. 

During the test, the activities were recorded using a camera and were labelled with time stamps for the activities to establish the ground truth. As shown in Figure 3, one AP and one receiver were used for this test: An AC1750 MU-MIMO gigabit router (Linksys, Irvine, CA, USA) was used for the AP and a Lenovo T400 laptop with Intel 5300 NIC was used for the receiver. The router provided a 3 × 3 multi-input and multi-output (MIMO) system using three built-in antennas. The router was configured to support the 802.11n AP mode at 5 GHz frequency. Internet control message protocol (ICMP) packets were transmitted at a sampling rate of 10 Hz (10 packets per second). Then, using a Linux CSI 802.11n tool [16], the CSI data were captured and extracted for 30 subcarriers for the first AP-Receiver antenna pair. The CSI amplitude data were then used to perform data processing for human-activity recognition.

### 4.2. Data Preprocessing

The raw CSI data contains high frequency noise, outliers, and artifacts introduced by rate adaptation rather than by human activities—high frequency noise manifests when the radios are switched to different modulation and coding schemes. Thus, a second-order, low-pass Butterworth filter was used to remove high frequency noise from the raw CSI data. The filter was configured to keep the sampling rate at 10 packets/s, and the cut-off frequency was set at 1 Hz. Although the walking activity causes higher Doppler shift than 1 Hz, the phase shift caused by occupant’s activity would be rotated due to hardware imperfection. Thus, the 1 Hz filtered CSI still contained the variation caused by walking activity. The variation in the filtered CSI was distinguished from other activities [15]. Figure 4 shows the raw and filtered amplitude data of one subcarrier. The filtered data became smooth after the high-frequency noise reduction.

### 4.3. Activity Segmentation

As shown in Figure 2, human activities cause signal changes, and the CSI variance has relatively high values when activities occur. If the CSI variances are greater than a threshold determined empirically, the CSI data may be considered to contain certain activities. While this approach somewhat successfully segments activities, when a human activity has small body movements or when some noise remains in the data after filtering has occurred, this approach will not always appropriately separate all activities, and the segmented data may additionally continue to contain unrelated data packets. 

As developing high-efficiency activity-segmentation algorithms is beyond the scope of this study, to increase recognition accuracy, the walking and in-place activities were manually segmented by referring to ground truth. A total of 400 data samples were segmented.

### 4.4. Activity Classification

Activity-classification approaches exploit various activity-recognition models to classify the segmented activity data. In this study, two different algorithms were used for activity classification: SVM and EMD reconstructed from literature [1,2,3,15,18,30]. An SVM model requires users to input features and their labels for training the model. In this case, the labels were extracted from the ground truth recorded via cameras, and the features were extracted from the input data—i.e., the segmented activity data. In the segmented activity data, the CSI data of the first data packet was represented as $H_{a}\left(s\right)$—with an $S_{b}\times 1$ dimensional vector—where $S_{b}$ indicates the 30 subcarriers, and s indicates the order of the $N_{a}$ successive data packets. The six characteristic features present in all subcarriers of the CSI amplitudes that were used for learning are: (1) the average of $H_{k}\left(s\right)$, (2) the standard deviation of $H_{k}\left(s\right)$, (3) the 25th percentile of $H_{k}\left(s\right)$, (4) the 75th percentile of $H_{k}\left(s\right)$, (5) the maximum of $H_{k}\left(s\right)$, and (6) the median absolute deviation of $H_{k}\left(s\right)$, where $\forall k\in \left[1,S_{b}\right]$ and $\forall s\in \left[1,N_{a}\right]$. 60 percent of the total segmented data were used for training, and the remaining 40 percent were used for testing. The SVM showed the anticipated labels for the testing data, so the labels resulting from the SVM were compared to the labels from the ground truth to determine model accuracy.

Unlike an SVM, an EMD finds the minimum cost of matching one distribution into another and thereby represents to what extent two distributions are similar to each other. Notably, under EMD, the same activity will have similar distributions across the CSI amplitudes, whereas different activities will have distinctive distributions. Figure 5 shows a histogram of the CSI distributions across all 30 amplitude subcarriers; here, “Bin” refers to the range of amplitudes, and “Amplitude Count” refers to the number of times the corresponding amplitudes appear in each Bin. In order to recognize activities, a known CSI-amplitude distribution for each activity needs to be selected; in this study, we selected three known distributions for walking, eating, and typing. Then, the EMD algorithm [33] was employed to calculate the EMD between the known distribution for the labeled activities and the unknown distribution for the unlabeled activities. If the EMD of an unknown activity manifested a minimal distance to one of the three known activity distributions, the unknown activity received a label for the corresponding activity.

## 5. Results

In this study, to determine the impact of environmental factors on the accuracy of two different CSI-based ADL-recognition systems, two subjects performed a walking activity and two in-place activities in different housing environments. To recognize activities, the authors used two different activity-recognizing approaches, an SVM-based model and an EMD-based model. The activity-recognition accuracy of the SVM-based model was 94.38% for Unit A and 87.50% for Unit B, whereas the EMD-based model showed 68.75% and 60.25% accuracy levels for Unit A and Unit B, respectively. Table 1 and Table 2 summarize the results from the SVM-based model and the EMD-based model, respectively. Both of the two ADL-detection algorithms show lower accuracies for Unit B than Unit A, which implies a mediating effect of environmental factors on the accuracy of the ADL-recognition systems.

Under the SVM-based model, features used for activity classification related to the amplitude of signal variance. Thus, even though an activity signal forms a specific waveform, its amplitude appeared to be low, so the SVM classifier considered the activity as no activity. The confusion matrix for Unit B showed 15 ‘no-activity’ data were predicted as examples of ‘typing activity,’ which mainly reduced the model’s accuracy. ‘Typing activity’ is one of the smallest movements in daily life, since when people type, they only move their fingers and arms a small amount. Thus, the signal variance of the ‘typing activity’ has a low amplitude, and if the signal power is reduced by environmental factors, the ‘typing activity’ signal may be too weak to differentiate from a ‘no activity’ signal. Accordingly, we expected—and observed—that Unit B environment would have a greater impact on signal propagation than Unit A environment. 

Under the EMD-based model, the distribution of the signals’ amplitudes has more impact on the activity classification than the amplitudes’ variance within the signal, since the EMD-based model classifies unknown activities based on the distributions’ similarity to known distributions. As shown in Figure 5, the amplitude of the ‘walking’ activity is evenly distributed over the bins, but the histograms of ‘eating’ and ‘typing’ activities show concentrated amplitude distributions in specific bin ranges. Thus, if the peak point of the amplitude distribution was shifted by changing the overall signal strength, the ‘eating’ and ‘typing’ activities could have similar amplitude distributions, which would mean that the EMD-based approach would consider those two different activities as the same activity. As shown in Table 2, the ‘walking’ activity in both Unit A and Unit B is well recognized, but the EMD-based approach achieved a low accuracy when predicting ‘eating’ and ‘typing’ activities. Table 2 also shows that many identified ‘no activity’ data were predicted to be ‘eating’ and ‘typing’ activities because the ‘no activity’ data manifested a stable signal wherein the signal strength was concentrated in a certain amplitude range. Thus, the histogram of the ‘no activity’ data appeared to be similar to those of ‘eating’ and ‘typing’ activities under the EDM approach.

The results also show that the EMD-based model has lower average accuracies compared to the SVM-based model. The SVM-based approach exploits 60% of the total 400 activities data to train the model and classified the remaining 40% activity data—160 samples. On the other hand, the EMD-based approach does not need training data, and consequently evaluates activity data for the full 400 samples. Thus, the actual number of recognized activities evaluated by the EMD-based model is greater than the number recognized by SVM-based model, which means the EMD-based model has more opportunities to make false predictions than the SVM-based model. Also, the SVM-based model used 60% of the total data as reference data, whereas the EMD-based model exploits only one reference datum for each activity to calculate the similarity of the known activity against the unknown activity data. Consequently and foreseeably, the EMD-based model does not cope with the variability of unknown activity data as well as the SVM-based model does.

To investigate the effect of time-dependency or subject-dependency of the result, we split data into each subject data and classified activities for each subject data; Each subject performed the experiment in different days. The results in different data segmentation showed that the accuracy of Unit A was generally higher than Unit B and the accuracy of SVM was higher than EMD, meaning that the time dependency and subject dependency did not have much impact on this result.

It is important to note that rather than identify effective ADL-recognition methods, this study compares the effect different environmental factors have on the performance of CSI-based ADL-recognition systems. Even though the SVM-based model and EMD-based model both show different accuracy levels for activity recognition, the results clearly indicate that both CSI-based ADL-recognition systems show less accuracy for the Unit B environment, meaning that the Unit B environment mediates the accuracy of the CSI-based ADL-recognition systems more than the Unit A environment does. We further discuss this important finding about the environmental effects mediating performance of the CSI-based ADL-recognition systems in the next section.

## 6. Discussion

### 6.1. Housing Environmental Factors

The results indicate that both of the tested CSI-based ADL-recognition models are quite sensitive to the difference of housing environment and yield significantly different accuracies between two units. Here, we mainly discuss the potential factors that affect the performance of the CSI-based ADL-recognition systems. While these factors may appear to be limitations to the present study, they serve—in fact—as a further defense of the justification of this study, since smart-home engineers must mediate these factors when designing tools within different environments.

#### 6.1.1. Building Materials and Household Items

One potential factor contributing the performance of ADL recognition is the difference of building materials. The main building materials of Unit B were reinforced concrete and concrete masonry, which is one of the hardest building materials for wireless signals to penetrate [23], while Unit A was built with predominantly plywood or drywall, which represent building materials that yield less signal loss. Such Wi-Fi resistant properties of concrete materials could potentially benefit the CSI-based ADL-recognition systems by blocking interferences from surrounding Wi-Fi networks and noises created from dynamic object movements outside of the unit, but it could inhibit the performance of CSI-based ADL-recognition systems, as the amplitude of transmitted signals becomes reduced during the signals’ travel indoors. This signal loss means that the signal variance caused by activities decreases, which challenges processes for differentiating noise, specific activities, and no activity. At least in our experiment, the disadvantage of having Wi-Fi resistant interior walls would have outweighed the advantage of having Wi-Fi resistant exterior walls and would have potentially contributed to the lower performance in Unit B.

The household item (e.g., furniture, appliance) is another potential factor for Wi-Fi signal propagation. Depending on the materials of household items, Wi-Fi signal is reflected from the object more or a lot of energy is absorbed. Unit B had more small and medium sized household items, which were not all made of metal, the more reflective objects compared to the wooden or plastic materials. Thus, the small and medium sized objects absorbed more energy of the Wi-Fi signal in Unit B than Unit A, which resulted in a reduced amplitude of the CSI. The decreased variation in the CSI, then, became hard to distinguish from no-activity or other activities in Unit B. 

#### 6.1.2. Population Density with Surrounding Wi-Fi Interference

The other contributing factor is the difference of population densities. Just as people can access home networks from outside their walls, people outside of a house can affect the Wi-Fi network environment inside. Thus, neighbors’ activities or outdoor movements will conceivably affect the performance of CSI-based ADL-recognition systems. Unit B had more nearby units than Unit A. Significantly, for Unit B, neighboring households shared walls on both sides of the apartment, whereas Unit A only had neighbors five meters away on one side of the testbed. Also, the elevator located next to Unit B continuously ran, which influenced the signals within Unit B; Unit A had no elevator. Thus, more movements unrelated to the test activities took place in proximity to Unit B, which could affect the received signals used for activity detection. Unit B was in a higher population apartment complex than Unit A testbed, meaning that there were more Wi-Fi networks around to influence the test. Thus, the surrounding Wi-Fi signals had a higher chance to interfere with the testing Wi-Fi signals in Unit B than Unit A. However, as mentioned in the previous section, such interference from surrounding Wi-Fi signals might have been alleviated by the building materials of Unit B, although Unit B had many openings.

### 6.2. Potential Strategies to Address Housing Environmental Factors 

In this section, we will discuss possible solutions for coping with the housing environmental effects influencing the accuracy of the CSI-based ADL-recognition system, including (1) using multiple receivers; (2) filtering external noise; and (3) using Wi-fi-friendly construction materials.

#### 6.2.1. Using Multiple Receivers

While our experimental setting used only one receiver, using additional receivers would improve the recognition accuracies of CSI-based systems by augmenting the variance of the received signals. In larger houses or those with multiple rooms, the strength of the signals arriving at the receivers becomes weaker, as the signals have to penetrate through multiple walls [10,23,24]. Under such circumstances, the reduced amplitude of the signal both becomes hard to distinguish from noise [3] and directly impacts the performance of the CSI-based ADL-recognition systems. For example, a receiver located in the living room may only capture reduced signal variance if an occupant performs an activity in the bedroom since the signal affected by the occupant must move through the environment—and walls—to arrive at the receiver in the living room; if the transmitter is not within the bedroom, the signal will lose even more energy as it travels from the transmitter to the bedroom and back to the receiver in the living room. However, if the environment includes an additional receiver in the bedroom, the affected signal will reach the receiver in the bedroom, thus reducing the additional energy loss. Having multiple receivers located in different rooms would therefore resolve the reduced-amplitude issue and improve accuracy. 

#### 6.2.2. Filtering External Noise

In order to address the effects resulting from a neighbor’s activity, advanced filtering methods must be proposed and used. While such filtering would represent a great challenge, developers may conceivably be able to distinguish the signal changes due to a neighbor’s activities in another unit, especially as such signals would have to penetrate multiple walls (possibly including exterior walls) and a neighbor’s activities may manifest different patterns in the frequency domain (e.g., different walking patterns). Lee et al. [34] demonstrated this possibility of differentiating simultaneous activities of multiple occupants using the frequency-domain features. Similarly, other researchers have successfully identified and removed specific subcarriers that were heavily influenced by a neighbor’s activities [3,35]. Such opportunities present options for researchers seeking to mitigate the effects of noise on CSI-based ADL recognition.

#### 6.2.3. Using Wi-Fi-Friendly Construction Materials

As smart homes become more popular, Wi-Fi-friendliness will play an important role in the design and selection of construction materials [36]. The performance of the CSI-based ADL-recognition systems relies on the magnitude of signal variance at the receiver, so the Wi-Fi-friendliness of building layout and materials will reasonably drive building designs seeking to harness CSI-based tools. Already, some options for improving Wi-fi-signal quality exist. For example, the interference from surrounding Wi-Fi networks could be alleviated by using the anti-Wi-Fi paint—which contains aluminum-iron oxide for absorbing high-frequency wireless signals [37]—on exterior walls. In addition, a porous wall that propagates Wi-Fi signals well [38] can be used for interior walls in order to reduce the attenuation of the signal that occurs as the signal passes through interior walls [38,39]. While such design choices may render the signal-resiliency goals underpinning CSI-based ADL recognition, there remains some uncertainty as to whether using such Wi-fi-friendly construction materials will actually improve performance, since CSI-based ADL recognition relies on the change of signals as the signals fade and reflect on surrounding walls. Further research will be necessary to examine how the use of such materials impact the performance of the CSI-based ADL recognition.

### 6.3. Limitations 

As this study intended to examine the effect of actual housing environments of existing homes on CSI-based ADL-recognition systems, various environmental factors were less controlled, and the results from such the experimental settings represent the combined effects of all possible environmental factors. Thus, the effect of each individual environmental factor is still unknown, although the experiment results confirm the significant effect of housing environment on CSI-based ADL-recognition systems. In addition, although two representative ADL-recognition algorithms were used in this study, the effect of housing environment could be largely dependent on algorithms.

## 7. Conclusions

The two robust CSI-based activity-recognition algorithms yielded lower accuracy in Unit B than Unit A, a result that speaks to the mediating influence of environment on CSI-based ADL-recognition systems. This result highlights that the effects of the housing environments should be considered when designing and implementing CSI-based ADL-recognition systems in various home environments. However, further research is necessary to analyze the isolated effects of various housing environmental factors and understand the impact of possible mitigation strategies on such environmental effects. 

## Figures and Tables

**Figure 1 sensors-19-00983-f001:**
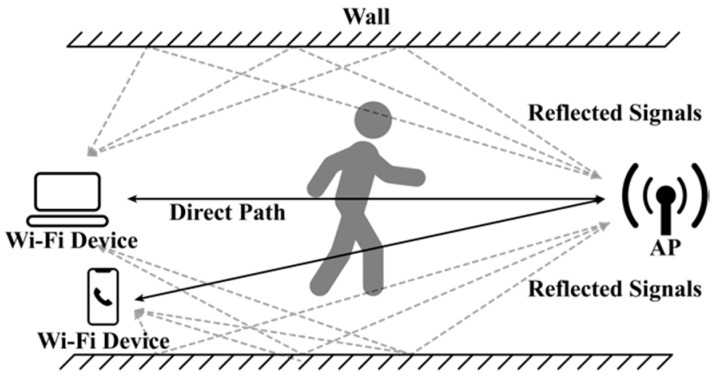
Multipath propagation of Wi-Fi signal indoors.

**Figure 2 sensors-19-00983-f002:**
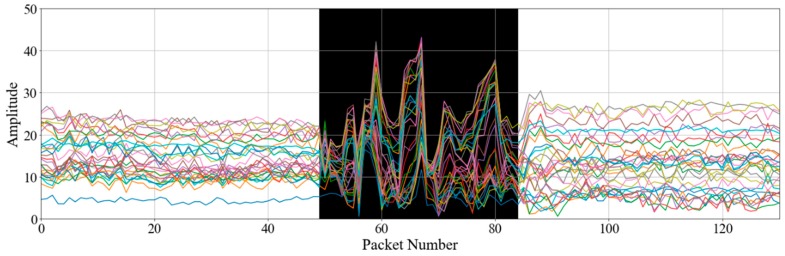
Raw CSI amplitude data of 30 subcarriers captured during walking. Colors indicate different subcarriers, and the black region indicates the walking activity.

**Figure 3 sensors-19-00983-f003:**
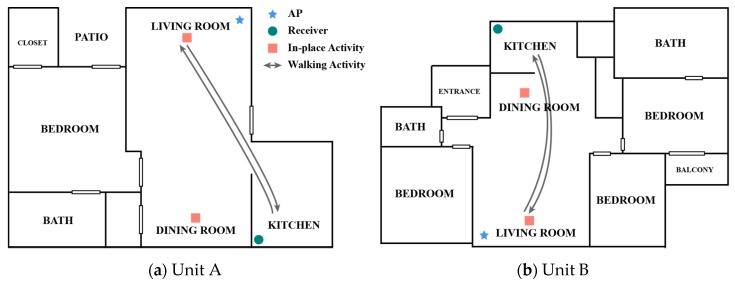
Floor plans for experimental test beds.

**Figure 4 sensors-19-00983-f004:**
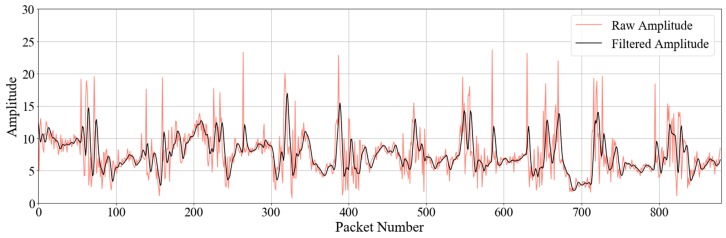
Comparison of raw and filtered amplitude data for one subcarrier.

**Figure 5 sensors-19-00983-f005:**
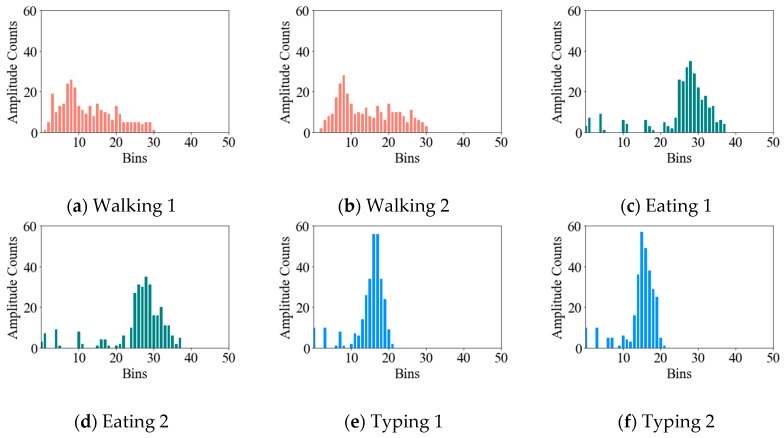
Amplitude counts in the amplitude bins for walking and in-place activities.

**Table 1 sensors-19-00983-t001:** Confusion matrix for Unit A and Unit B using SVM-based model (Data shown here represent 40% of garnered data).

	Unit A				Unit B			
	Walking	Eating	Typing	No Activity	Walking	Eating	Typing	No Activity
Walking	38	0	0	0	38	0	0	0
Eating	3	35	0	0	0	38	0	0
Typing	2	0	31	0	0	1	30	2
No Activity	4	0	0	47	2	0	15	34

**Table 2 sensors-19-00983-t002:** Confusion matrix for Unit A and Unit B using EMD-based model.

	Unit A				Unit B			
	Walking	Eating	Typing	No Activity	Walking	Eating	Typing	No Activity
Walking	96	4	0	0	92	0	0	8
Eating	0	73	2	15	0	28	21	41
Typing	0	21	46	23	0	28	62	0
No Activity	0	26	34	60	0	24	37	59

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
