# Peer review of "The Effects of Housing Environments on the Performance of Activity-Recognition Systems Using Wi-Fi Channel State Information: An Exploratory Study"

_sensors, 2019, doi:10.3390/s19050983_

Round 1

Reviewer 1 Report

The paper is well written and very interesting. It explores the effect of different type of housing environment on two algorithms for ADL recognition. The theme of the influence of different environments over RF-based algorithms for localization and, more in general, activity recognition is of utmost importance and very interesting.

The methods, discussion and results are well described and a proper introduction with reference to previous work are correctly reported.

However, I have a couple of observations: for the experiments, it seems that only two individuals have performed the tests. Did they both perform the tests in both the environments? How many times they repeated the experiment? In my experience, due to time-dependent nonidealities, the same experiment repeated by the same user in two different times could give different results. The authors reported the users’ heights and not their weights, that is in my opinion an important parameter in this kind of experiments. Why in Table 1 are reported only the 40% of the gathered data? The data should be enlarged to give them statistical soundness.

Author Response

Comment 1

The authors reported the users’ heights and not their weights, that is in my opinion an important parameter in this kind of experiments.

Response

Thank you for reminding us how important it is to present detailed physical information of the subjects. We agree that the weight as well as height of subject is important parameters for our test.

Changes (line: 231-234)

Two subjects were recruited for the experiments at each Unit; Recruiting for the experiment at each Unit were conducted separately due to the geographical distance. All the four subjects participating in both Units were male having similar physical characteristics; Their heights ranged from 4375px to 180cm, while their weights range from 68kg to 70kg.

Comment 2

Did they both perform the tests in both the environments?

Response

Thank you for this excellent observation. Since the two tests were conducted in different countries, the identical subjects could not all proceed with the test. Instead, we first performed the test with two subjects in the United States, and then we recruited two subjects who had the similar physical characteristics as the subjects of the test in the US for the second test in the South Korea.  

Changes (line: 231-234)

Two subjects were recruited for the experiments at each Unit; Recruiting for the experiment at each Unit were conducted separately due to the geographical distance. All the four subjects participating in both Units were male having similar physical characteristics; Their heights ranged from 4375px to 180cm, while their weights range from 68kg to 70kg.

Comment 3

How many times they repeated the experiment? In my experience, due to time-dependent nonidealities, the same experiment repeated by the same user in two different times could give different results.

Response

Each subject repeated the experiment 10 times, and the second subject performed test the day after the first test. While time-dependent nonidealities were observed in terms of the overall amplitude levels, variances caused by subjects’ activities were clearly distinguishable in any dataset collected in different times. We have split data in a way to reflect time dependency or subject dependency, but the difference of the overall accuracy in two testbeds were consistent in any data segmentation.

Changes (line: 355-359)

To investigate the effect of time-dependency or subject-dependency of the result, we split data into each subject data and classified activities for each subject data; Each subject performed the experiment in different days. The results in different data segmentation showed that the accuracy of Unit A was generally higher than Unit B and the accuracy of SVM was higher than EMD, meaning that the time dependency and subject dependency did not have much impact on this result.

Comment 4

Why in Table 1 are reported only the 40% of the gathered data? The data should be enlarged to give them statistical soundness.

Response

Thank you for your assessment. We used 60% of the total segmented data for training, and the remaining 40% were used for recognizing subjects’ activities. Thus, Table 1 reports only the 40% of the gathered data which the SVM-based model predicted. Wi-Chase, a reference paper, used 50% of total data for training. On the other hand, Table 2 shows 100% of the data since EMD-based model calculates the similarity of all segmented data. This study examines the effect of differences in the housing environment between Unit A and Unit B on the accuracy of the two classifiers rather than comparing what classifier is better. Then, the two different classifiers produce better results on Unit A than B.

Reviewer 2 Report

The paper studies the impact of different housing environments on the performance of RF-based activity recognition systems, in particular off-the-shelf NIC devices. This topic has been rarely addressed in the literature, though it is of great importance for the design of smart homes and IOT healthcare devices. The paper is well-written and reviews the existing literature in a concise manner. There is a major issue and some minor issues that I would like to ask the author to address:

Major Comment:  

The maximum Doppler shift is given by (v0*f0)/c0, in which v0 is the walking speed, f0 is the operating frequency and c0 is the speed of light. Roughly speaking, at 5 GHz, this turns to v0*5e9/3e8, meaning 16.6*v_0. If the walking speed is (reasonably) around 1 m/s, the maximum Doppler shift becomes around 16 Hz. Other Doppler frequency components might be smaller due to the angles, etc. 

Then the immediate question is that if you have filtered your data with a LPF of 1 Hz cut-off frequency (according to Sec. 4.2), you have indeed filtered most of the frequency components caused by the walking activity. 

Please elaborate on this issue. 

Minor Comments: 

The experimental setup should also be explained from the interior and the type/number of furniture point of view. Please add some notes about it.  

Section 6 can also benefit from some discussions about the impact of static objects and the richness of in-home scatterers perspective. 

Sec. 4.1, lines 225-227: it reads as if there have been two persons walking in a room. Please modify the sentence to avoid any misunderstanding. 

Sec. 3, line 188: use math-form for mathematical notations. Please apply everywhere in the manuscript. 

Round 2

Reviewer 1 Report

The authors address all my comments. In my opinion, the paper can be accepted now.

Author Response

We are glad to hear that the reviewer found all the issues were resolved.

Reviewer 2 Report

The authors have addressed my comments appropriately. Concerning my major comment, I trust the authors' observations and am satisfied with the additional clarifications in the paper, but I am less satisfied with the scientific reasoning of the observation. I think the main reason for you to be able to capture the walking activity even after such a LPF is the hardware-based phase shift (undesired shift). Probably, the desired phase shift caused by the walking activity is rotated (due to hardware imperfection) such that the final phase allows for Doppler frequencies below/near 1 Hz. Therefore, it still carries the fingerprints of the walking activity. If you agree, you might add some more line to the manuscript about this issue. Otherwise, it is alright and I am not gonna stop the publication process. 

A small update on the existing literature: the following papers are among the newest in the field and might be relevant to be reviewed in the manuscript:  

Borhani et al., â€śA non-stationary channel model for the development of non-wearable radio fall detection systems,” IEEE TWC

Nopphon et al. "Mitigation of CSI temporal phase rotation with B2B calibration method for fine-grained motion detection analysis on commodity Wi-Fi devices", Sensors
